# Developing a Health-Promotion Program Based on the Action Research Paradigm to Reduce Cardiovascular Disease Risk Factors among Blue Collar Workers

**DOI:** 10.3390/ijerph16244958

**Published:** 2019-12-06

**Authors:** Won Ju Hwang, Jin Ah Kim

**Affiliations:** 1College of Nursing Science, East-west Nursing Institute, Kyung Hee University, 26 Kyungheedae-ro, Dongdaemun-gu, Seoul 02447, Korea; hwangwj@khu.ac.kr; 2Department of Nursing, Gyeongju University, 188, Taejong-ro, Gyeongju-si, Gyeongsangbuk-do 38065, Korea

**Keywords:** workplace, occupational health, cardiovascular disease, participatory action research, health promotion

## Abstract

This study developed and evaluated a health management program based on the participant-centered concept of action research to reduce cardiovascular disease (CVD) risk factors among blue collar workers. Data from structured questionnaires completed by 32 workers in a small-to-medium sized workplace from September 2015 to October 2016 as well as participants’ anthropometrical (weight and waist) and biological (blood pressure, glucose, total cholesterol, triglyceride, high-density lipoprotein (HDL), and low-density lipoprotein (LDL) cholesterol) data were analyzed using paired t-test and Fisher’s exact test. To examine the longitudinal effect of the intervention, survival analysis and linear mixed model (LMM) were used. There was an improvement in participants’ self-regulation in maintaining health-promoting behaviors, body weight, blood pressure, and HDL cholesterol following the intervention. Furthermore, the effects of the health management program continued even after the program ended. These findings suggest that the health management program developed in this study could be effective in reducing CVD risk factors among workers in small-to-medium sized workplaces and should be applied to other small-to-medium sized workplaces to foster health-promoting behaviors.

## 1. Introduction

Cardiovascular disease (CVD) accounts for 12.8% of all causes of deaths worldwide and is the leading single cause of deaths [1]. In Korea, CVD mortality rate also consistently increased over the last decade, indicating the need to manage the disease more efficiently. CVD mortality rate in Korean workers, in particular, increased from 37.5% in 2014 to 38.4% in 2015 [2], warranting the need for public health efforts to reduce CVD among workers in Korea. Moreover, small-to-medium sized workplaces, which employ 85% of all workers in Korea, are not required to have an in-house occupational health manager, and there is no legal provision mandating this for small businesses with fewer than 50 employees [3]. Consequently, the health of workers in small-to-medium sized workplaces is not systematically managed by professional occupational health personnel. A previous study reported that CVD mortality rate in employees in small-to-medium workplaces is higher than those in large-scale workplaces [4]. Among all health issues in workers, the lack of CVD management, in particular, requires urgent attention. Therefore, many intervention studies have been published to address these problems. However, researches have shown that health programs developed by researchers induce short-term, temporary changes in knowledge and behavior [5]. One reason is that the health-promotion programs for improving public health have been mainly developed in researcher-centered studies based on a natural science paradigm utilizing standardized objective scales, with the purposes of identifying precise laws to explain, predict, and control a phenomenon and of applying and generalizing these laws [6]. Another reason for the lack of long-term effects of such programs is that researchers are concerned with obtaining generalizable academic knowledge rather than with solving real-life problems and making changes [6]. Consequently, many intervention studies are limited in explaining and solving a problem in a real-life situation, in which a dynamic causal relationship is at work, and these interventions only demonstrate short-term effects rather than long-term effects [7].

Contrary to the researcher-centered method in an up-to-down direction, wherein the researcher identifies a problem, develops a program to solve the problem, and applies the program to the research participants, the participant-centered method in a down-to-up direction, wherein programs are developed and applied as participants identify their health problems and ways to address it with the help of the research, has been recognized as an effective intervention approach because it can handle real-life problems more efficiently and proactively [7]. In participant-centered research, participants go through a cycle of making a plan, of observing the process while executing the plan, of reflecting on the outcome, of revising the plan, and of collecting additional data, making this research process itself a real-life application. Thus, this approach is suitable for an intervention-based study of CVD, which requires not a one-time but continuous intervention. Additionally, participant-centered research improves health-related self-regulation and health-promoting behavior (psychological and behavioral control at a personal level to achieve self-decided goals) because participants can control their own thoughts, actions, and emotions unlike those in researcher-centered studies. Health problems which are closely associated with lifestyle habits, such as CVD, can be prevented and managed only by improving lifestyle habits and by sustaining desirable habits and health behaviors in the long term [8]. Thus, rather than a high-impact, short-term intervention, there is a need to develop a long-term, cost-effective intervention program that does not demand considerable time and effort even if the impact is low. Therefore, a participant-centered intervention study is considered to be an effective way to address health problems in employees in small-to-medium sized workplaces.

Therefore, the purpose of the present study was to help workers identify their health problems by themselves, to seek ways to address the problem, and to put those solutions into practice. The study aimed to decrease CVD risk and to ultimately examine the effectiveness of the participant-centered health-promotion program in improving and sustaining workers’ self-regulation and health-promoting behavior.

## 2. Study Purpose

The purpose of this study was to develop and apply a participant-centered health-promotion program based on the concept of participatory action research to reduce CVD risk factors in workers in small-to-medium sized workplaces. To do so, the participants in this study explored CVD risk factors with the researcher and completed a process which involved making plans to address the risk factors, putting the plans into practice, observing behavioral changes, and reflecting on the health behaviors and the process of change. The primary aim of this study was to statistically confirm the potential of the program as an effective intervention approach to prevent and manage CVD risk factors and its effect on sustaining health-promoting behavior in workers in small-to-medium sized workplaces.

## 3. Methods

### 3.1. Study Design

This intervention study developed and applied a participant-centered program to reduce CVD risk in workers in small-to-medium sized workplaces without an in-house occupational health manager. The study duration was approximately 13 months, from October 2015 through October 2016.

### 3.2. Conceptual Framework

The conceptual framework of this study was the participatory action research methodology proposed by Lewin [9], which involves five stages. First, the characteristics of a problem in a field are understood and reconceptualized. Second, action plans are established based on these characteristics. Third, the plans are put into action. Fourth, behavioral changes are observed and unintended behavioral changes and problems occurring due to these changes are immediately addressed. Any problems that might be accentuated while executing the plans are also identified in this stage. The final stage is reflection, which involves reviewing the outcomes of the plans, identifying problems that have yet to be resolved, determining the future direction of the research, and providing basic data for further participatory action research. These stages are repeated until the participants achieve their health goals and observe an improvement in their self-regulation ability to maintain health-promoting behavior (Figure 1).

### 3.3. Participant

A small-to-medium sized business enterprise was selected. The company developed and manufactured acoustic panels and speakers. According to the 2015 Periodic Health Examination at workplaces, 41.8% of the employees in this workplace were assessed to have a moderate-to-severe risk for CVD. Based on this finding, 49 workers (30 assessed to have moderate CVD risk and 19 assessed with a high risk) were selected as potential study participants. Subsequently, after obtaining permission from the business owner, a recruitment announcement for the study was made at the workplace with the help of an in-house occupational health manager and a briefing with the 49 potential participants was held to fully inform them of the study. Of the 49 individuals, 34 volunteered to participate (19 with moderate CVD risk and 13 with high CVD risk). However, during the study period, two participants left the company, and the final number of participants was 32. The CVD risk groups were determined based on the CVD risk assessment criteria established by the Korea Occupational Safety and Health Agency (KOSHA) based on the 2003 WHO International Society of Hypertension guidelines (WHO-ISH) [10] (Figure 2).

### 3.4. Ethical Considerations

Participants were informed of the study objectives and procedure in detail during the recruitment process. They were assured that the collected data would not be used for any other purpose other than the present research study and that the raw data would be destroyed after study completion. They were also informed that they could withdraw participation at any point without penalty, that they would remain anonymous in all reports of the study, and that any personal information including names would never be revealed. All participants provided written informed consent and received a copy of the signed consent form and the study protocol. Anonymity was guaranteed by assigning an allocation number (AN) to the data and by storing them in a locked cabinet. The study protocol was approved by the Institutional Review Board of K University (IRB No. KHSIRB-15-023).

### 3.5. Data Collection

The instruments used in this study are as follows (Table 1).

#### 3.5.1. Self-Regulation

To assess participants’ self-regulation ability, that is, self-regulation for exercise, diet, smoking, and drinking, the Treatment Self-Regulation Questionnaire (TSRQ) [11] was used. The TSRQ consists of 15 items across 3 subscales: autonomous motivation (6 items), externally controlled motivation (6 items), and amotivation (3 items). Each of the subscales can be used independently. The present study only assessed the level of improvement in autonomous motivation, using 24 items on autonomous motivation for exercise, diet, smoking, and drinking. Higher TSRQ scores indicate a higher level of autonomous motivation. The reliability coefficients measured by Cronbach’s α were 0.81, 0.79, 0.84, and 0.89 for exercise, drinking, smoking, and diet, respectively.

#### 3.5.2. CVD Risk Perception

CVD risk perception was assessed using an instrument developed by Becker and Levine [12], which was translated and validated for the Korean population [12]. The scores range from 4 to 20, with higher scores indicating higher CVD risk. The Cronbach’s α of the instrument was 0.80 in the original study and 0.72 in the present study.

#### 3.5.3. Health-Promoting Behavior

Health-promoting behavior was assessed using the Health-Promoting Lifestyle Profile II (HPLP II) [13], an instrument developed by Walker, Sechrist, and Pender. The scale was translated into Korean by Hwang [10]. The HPLP II consists of 52 items, and the higher the score, the higher the level of health-promoting behavior. The Cronbach’s α of the instrument was 0.94 at the time of development. The Cronbach’s α values in the present study were 0.92, 0.89, 0.93, 0.79, 0.80, and 0.86 for health responsibility, physical activity, nutrition, spiritual growth, interpersonal relation, and stress management, respectively. The Cronbach’s α of the overall scale was 0.88.

#### 3.5.4. Family Function

Family function was assessed using the Family APGAR Questionnaire developed by Smilkstein [14]. The instrument consists of five items (adaptation, partnership, growth, affection, and resolve), which are rated on a 3-point scale. The total score ranges from 0 to 10, and higher scores indicate better family function. Scores 7–10 are interpreted as a highly functional family, scores 4–6 are interpreted as a moderately functional family, and scores 0–3 are interpreted as a severely dysfunctional family. The Cronbach’s α of the instrument was 0.72 in the original study and 0.80 in the present study.

#### 3.5.5. Interpersonal Support Evaluation List-12 (ISEL-12)

Perceived social support from family, friends, and coworkers was assessed using the Interpersonal Support Evaluation List (ISEL-12), developed by Cohen, Mermelstein, Kamarck, and Hoberman [15]. For use in the present study, the ISEL-12 was translated into Korean and back-translated, and the translation versions were reviewed by a nursing professor. The reliability of the Korean version of the instrument was evaluated based on the data obtained from 132 workers. The ISEL-12 consists of 12 items, each rated on a 4-point scale (definitely false, probably false, probably true, and definitely true). The scores range from 0 to 36, with higher scores representing higher levels of perceived social support. The Cronbach’s α was 0.79 in the original study, and its content validity was tested by three experts in this area. The Cronbach’s α of the Korean version was 0.81 in the pilot study conducted with 132 workers and 0.86 in the present study.

#### 3.5.6. Job Stress

Job stress was assessed using the Effort-Reward Imbalance (ERI) questionnaire developed by Siegrist [16]. The questionnaire is based on the effort-reward imbalance model and consists of 23 items (6 items on effort, 11 items on reward, and 6 items on overcommitment). Each item is rated on a 5-point scale from 1 to 5, with 30 and 55 being the best possible scores for effort and reward, respectively. Job stress is assessed using the ER ratio, obtained by dividing the total effort score by the total reward score. Specifically, the ER ratio is calculated by first adjusting the total reward score with a correction factor, 6/11 (= 0.545) to transform the total scores of effort and reward to 1 (as there are 6 and 11 items in each of the subscales, respectively), and by computing the ratio with the transformed reward score as the denominator and the effort score as the numerator. An ER ratio less than 1 indicates low-level job stress, and an ER ratio greater 1 indicated the presence of high-level job stress. The Cronbach’s α values in the original study were 0.78 and 0.81 for effort and reward, respectively, and its construct validity was confirmed by performing a confirmatory factor analysis [16]. In the present study, the Cronbach’s α values for effort and reward were 0.79 and 0.77, respectively.

#### 3.5.7. Anthropometric Measurements

Height and weight were measured using an electronic measuring scale, and the body mass index (BMI) was computed. Waist circumference was measured around the ilium using a measuring tape. Blood pressure was measured using a manual sphygmomanometer after a 15-min resting period.

#### 3.5.8. Blood Test

Pre-intervention data on high-density (HDL), low-density lipoprotein (LDL), and total cholesterol; triglycerides; and fasting blood glucose were obtained from the blood tests performed in September 2015 by the Department of Occupational and Environmental Medicine at K University Hospital as part of the periodic health examination at workplaces. Post-intervention data used to evaluate hematological changes following the application of the participant-centered health-promotion program were obtained from the blood tests performed in September 2016 by the Department of Occupational and Environmental Medicine at K University Hospital as part of the periodic health examination at workplaces.

### 3.6. Contents and Process of Action Research

#### 3.6.1. Action Research—Development of a Health Management Journal and Testing its Content Validity

To develop a health management journal based on the cyclic nature of action research [9], we conducted a systematic review of the literature, including action research studies based on critical theory, studies conducted to develop intervention programs to prevent and manage CVD in workers, and action research-based intervention studies conducted with workers. Additionally, two nursing professors were consulted. Based on the literature review and the input from the nursing professors, a health management journal for the prevention of CVD was developed and reviewed by an occupational nurse. The journal consisted of the following sections: “thinking of health problems” to explore CVD-related problems, “establishing health goals and making action plans”, “reflecting”, and “reestablishing health goals and making action plans accordingly”. To examine the content validity of the health management journal, two nursing professors and two occupational nurses were consulted. The journal was revised based on their opinions along with the suggestions of an expert of health management intervention programs, and its final contents and design were determined.

#### 3.6.2. Participants’ Baseline Characteristics

Data on participants’ baseline characteristics before the intervention were collected in November 2015 by conducting a focus group interview and survey and by measuring participants’ weight, blood pressure, and waist circumference. Additionally, blood test results (HDL/LDL/total cholesterol, triglycerides, and fasting blood glucose) from the Periodic Health Examination at workplaces performed in September 2015 were obtained.

#### 3.6.3. Using the Health Management Journal

Thinking of health problems: The health management journal begins with “thinking of health problems” as the first step, which corresponds to the first stage of action research—reconceptualization. During the interviews, the researcher focused on stimulating the participants to identify their own CVD risk factors. To that end, participants were provided an individualized CVD risk assessment table created by the Department of Occupational and Environmental Medicine at K University Hospital based on the results of the 2015 Periodic Health Examination at workplaces. The researcher explained the results in detail, if requested. Participants reviewed the risk factors (e.g., smoking, physical activity, BMI, etc.) in the table and were stimulated to think of specific causes and reasons for not being able to modify the undesirable health habits and to write them in the “thinking of health problems” section of the health management journal.

Establishing health goals and making action plans: This step corresponds to the planning stage of the cyclic structure of action research. Participants were instructed to set long-term (12 months) and short-terms (2 months) goals to address their health problems identified in the previous stage. Participants were encouraged to set specific goals that they could achieve. They were then instructed to make practical action plans (regardless of the impact of the goal) and to record them in the “making action plans” section of the health management journal.

Putting action plans into practice: Participants were instructed to put their action plans into practice while contemplating their perceived health problems and the health goals they established for themselves until their next meeting with the researcher two months later. To prepare for the next stage, they were also instructed to observe and try to remember how they put the plans into practice and to occasionally make notes in the journal regarding health behavior changes, new problems, and any new resources they needed. To stimulate participants to frequently contemplate their plans and to help them adhere to it during the two-month practice period, the researcher sent a text message on their mobile phone once every other week and played the role of an aide to the participants during the course of the study.

Observing: This stage began two months later, when the researcher and participants met again in February 2016. In this stage, participants reviewed the process of putting their plans into action and explored any changes in health behavior, new problems in the process, and resources they needed to put the plans into practice.

Reflecting (confirming whether the health action plans were executed and reflecting on the outcomes): In this final stage of the cyclic action research process, health action plans were executed and participants reflected on the outcomes. Participants were instructed to assess which of the plans were executed well and which were not, to contemplate on the poorly-executed plans to understand why they were not executed well, and to record their own evaluation and reasons in the health management journal.

Repeating the cyclic action research process: The cyclic nature of the action research process enhanced participants’ self-regulation ability with respect to their health management behavior, as it involved iterating the cycle for a total of four times during the research periods (Figure 1).

#### 3.6.4. Post-Intervention

In September 2016, participants’ post-intervention data were collected using the survey and anthropometric measures, which were used to assess participants’ baseline characteristics. Participants’ blood test results (HDL/LDL/total cholesterol, triglycerides, and fasting blood glucose) were obtained from the 2016 Periodic Health Examination at workplaces conducted in the same month. Among the measures included in the survey, the HPLP II was administered for a total of four times with a three-month interval to examine the sustainability of the health-promoting behavior.

### 3.7. Data Analysis

Data were analyzed using SPSS/WIN 22.0 (SPSS Inc., an IBM Company, Seoul, Korea), and the significance level was set to 0.05. Participants’ general and disease-related characteristics were analyzed using descriptive statistics and presented as means and standard deviations. To investigate the effectiveness of the participant-centered health-promotion program, Fisher’s exact test and paired t-test were performed to compare the pre- and post-intervention survey data, anthropometric measurements, and blood test results.

Kaplan-Meier survival analysis, a method used to estimate the cumulative survival rate by computing an interval survival rate at each point of an event occurrence while maximizing the use of partial information in truncated data, was used to predict whether health behavior could be maintained in participants after the completion of the participatory action research. To collect the data required for survival analysis, the HPLP II was administered at months 0, 4, 7, and 10 following the initiation of the action research process. If a participant’s HPLP II score at a time point was higher than the score in the preceding time point, it was determined that the health-promoting behavior had improved. Participants whose HPLP II scores continuously increased throughout the study period were determined to have maintained their health-promoting behavior. If a participant’s HPLP II score at a time point was lower than the score at the preceding time point, it was determined that the participant had stopped engaging in health-promoting behavior. Based on this criterion, data from 8 participants were considered as truncated; thus, data from a total of 10 participants, including 2 who had left the company, were considered as truncated data. To identify the factors influencing the sustainability of CVD-related health-promoting behavior in participants, Cox regression analysis was performed. This method can identify the predictors of sustainability by controlling for the confounding variables. The analysis model was constructed by taking into account the theoretical considerations and the variables found to be significant in the univariate analysis. The following variables were included in Cox regression analysis: CVD risk perception, knowledge of CVD, self-regulation, social support and family relationship, and job stress. Survival analysis using a Cox model is based on the proportional hazards assumption. Accordingly, it should first be confirmed that the assumption is satisfied. A visual inspection using a plot is an easy method to test whether the assumption is satisfied, while a more objective method is to use test statistics on time-dependent variables. In the present study, the proportional hazards assumption was first tested using a log-minus-log (LML) plot and then by analyzing the interaction between CVD risk and time.

Additionally, a linear mixed model (LMM) was used for the repeated measures of the HPLP II to examine any changes in participants’ CVD-related health-promoting behavior over the 13 months and to predict future behavioral change based on pattern analysis of the health-promoting behavior.

## 4. Results

### 4.1. Demographic Characteristics of the Study Participants

Most of the participants were male (81.2%), and 65.5% of them were in their 40s or 50s. More than half (56.2%) of the participants had a high school or lower level of education, and 62.5% earned no more than 2 million won per month. The average number of work hours per week was 55.6, which is longer than the legal limit of 40 h per week. Most participants were employed full time (84.4%). Participants’ mean work experience was 5.2 years, and 56.2% of participants had a work experience of 5 year or more. More than half (56.2%) of the participants were smokers. Half of the participants (50.0%) reported a family history of CVD, such as stroke, hypertension, and diabetes. Based on the KOSHA risk assessment criteria (2009) applied to the data from the 2015 Periodic Health Examination at workplaces, 59.3% of participants were categorized into the moderate-risk group and 40.6% were categorized into the high-risk group (Table 2).

### 4.2. The Effects of the Action Research Program

The comparison of the pre- and post-intervention data revealed the following results. HDL cholesterol increased from 55 mg/dL at baseline to 62 mg/dL post-intervention (*p* < 0.001). Improvements in LDL cholesterol, total cholesterol, and fasting blood glucose were observed, although these were not statistically significant. Regarding anthropometric measurements, participants’ weight decreased post-intervention (*p* = 0.006), and the proportion of participants classified as obese based on their BMI index at baseline reduced from 53.2% to 37.5% post-intervention (*p* < 0.001). Systolic and diastolic blood pressure also decreased post-intervention (*p* < 0.001 and *p* = 0.037, respectively). In addition, CVD risk perception decreased (*p* = 0.003) and health-promoting behavior increased (*p* < 0.001) post-intervention. Among the self-regulation abilities, self-regulation of exercise (*p* = 0.002), diet (*p* = 0.001), and smoking (*p* < 0.001) had increased. Additionally, family function post-intervention had increased (*p* = 0.001); however, job stress further increased (*p* = 0.003), and no difference in social support was observed. The proportion of current smokers post-intervention decreased from 56.3% to 34.4% post-intervention. As the participants classified as high-CVD-risk based on the 2015 data of the Periodic Health Examination at workplaces were reclassified as low or moderate CVD risk after participating in the action research, the proportion of participants in the high-CVD-risk group significantly decreased from 40.6% to 21.9% (*p* = 0.008; Table 3).

Linear mixed model was used to find out the intervention effect. Regarding the changes in participants’ health-promoting behavior over the course of intervention, 40.6% of them engaged in more health-promoting behaviors at month 4 compared to that at baseline. Further, 65.6% of them engaged in more health-promoting behaviors at month 7 compared to that at month 4 and 75.0% of them engaged in more health-promoting behaviors at month 10 compared to that at month 7. Thus, 75% of participants maintained health-promoting behaviors in the final month of the action research process (month 10). Also, the spaghetti plot showed increasing tendency of health-promotion behavior. The comparison of the mean HPLP II scores at months 0, 4, 7, and 10 showed that health-promoting behavior among participants had increased over time (Figure 3).

The Kaplan-Meier survival analysis was performed to predict whether participants would continue practicing the health-promoting behavior even after the completion of the program. The result showed that participants were likely to continue engaging in health-promoting behavior (Figure 4).

The Cox regression analysis predicted that the rate of the health-promoting behavior continuity would increase by 4.0 times, as CVD risk perception increases by a unit, after controlling for all other covariates (*p* = 0.016), and that it would increase by 1.05 times as self-regulation ability increases by a unit after controlling for all other covariates (*p* = 0.002; Table 4).

## 5. Discussion

### 5.1. General Characteristics

The mean age of the 32 study participants was 42.5 years, and a large proportion of them were in their 40s or 50s. Based on previous studies, which identified age as a risk factor for CVD, the mean age of the participants is high because the present study targeted workers with moderate to high CVD risk based on the Periodic Health Examination at workplaces. Generally, employees of small-to-medium sized businesses tend to be older than those of large-scale businesses, and blue-collar workers tend to be older than office workers [17]. Therefore, considering the general attributes of workers in small-to-medium workplaces, who are on average older than those working in medium-to-large sized workplaces and office workers, public health professionals should recognize that workers in small-to-medium sized workplaces are at a high risk of CVD, which requires timely attention and risk management. Additional CVD risk factors include low socioeconomic status [18] and working overtime and working in shifts [19]. In the present study, 56.3% of the participants had an education level of high school or lower level and 62.7% had a monthly income of no more than 2 million won. Furthermore, the mean number of work hours was 55.6 per week, exceeding the legal limit by 15.6 h. These findings are in line with a previous study’s findings that low socioeconomic status and working overtime are CVD risk factors [18,19]. Considering the general attributes of employees in small-to-medium sized workplaces and their work environment (such as working overtime), this group can be categorized into a vulnerable group at risk for CVD. Accordingly, to prevent the occurrence of CVD in employees of small-to-medium sized workplaces who work overtime, financial support at an organizational or policy level should be provided to promote workers’ health and it is imperative to reduce the number of work hours through reasonable negotiation.

### 5.2. Biological and Anthropometric Measurements

The analyses of the blood tests and anthropometric measurements pre- and post-intervention showed that the cholesterol and fasting blood glucose levels improved and, in particular, HDL cholesterol increased significantly. Further, weight and blood pressure had significantly decreased. A participant-centered action research study of health management of male blue-collar construction workers in Australia [18] reported that, despite the workers’ high level of interest in maintaining a healthy diet and exercise regimen, there was still no change in the rate of health-promoting behavior after the completion of the study due to excessive work burden, lack of time, and fatigue. Accordingly, the researchers emphasized the need for an individualized approach [18]. In comparison, the present study observed an increased rate of maintaining health-promoting behavior because it focused on helping participants resolve their own health problems and used an approach tailored to participants’ situation, which enabled them to establish specific health management plans. The study findings indicate that using a personalized approach using the action research paradigm to stimulate workers to establish and execute practical plans on a regular basis might be effective in reducing CVD risk.

### 5.3. Changes in Health-Promotion Behavior

To examine actual change in health-promoting behavior as a result of the participant-centered health-promotion program, HPLP II was used to measure the level of change in individual workers’ health-promoting behavior. Additionally, the level of improvement in self-regulation of health behavior was assessed to indirectly observe the sustainability of the health management program. The results showed that health-promoting behavior increased after the intervention compared to before intervention. Moreover, participants’ health-related self-regulation ability improved as they engaged in health behavior. In a previous study of male, blue-collar construction workers [19], health-promoting behavior of the participants was assessed for both physical and psychological health and the findings showed that the behavior improved through action research, which is consistent with the present study’s findings. Self-regulation has been stressed as a critical element when individuals proactively modify their own behavior or change the external environment to achieve their health management goal and to maintain their health-promoting behavior [11,20]. Self-regulation of health behavior is an important factor for goal setting and goal striving [21]. This study showed that participant-centered research is an effective approach in increasing self-regulation of health-promoting behavior by stimulating participants to be proactive and by repeatedly applying the cyclic process of reconceptualization, planning, putting the plans into practice, observing, and reflecting. However, a previous study suggested that community-based participatory action research is effective in the continuous and systematic management of chronic disease [20]. Moreover, according to the previous study, it is necessary to consider not only individual factors but also organizational factors when planning a CVD risk reduction program. [22,23]. Thus, changes at the organizational level (such as the addition of health-monitoring tools including a sphygmomanometer and weighing scale, the provision of financial support for exercise, the revision of the menus at workplace cafeterias, and the enforcement of nonsmoking policies at workplaces), community support, and national policy should also be made to manage workers’ health.

### 5.4. Changes in Psychosocial Factors

Family function, as perceived by the study participants, had improved post-intervention, although the program had little effect on job stress and perceived social support. A previous study identified psychosocial work environment (i.e., high job demands and low job latitude) as a factor of job stress [24,25]. Therefore, in order to reduce job stress, it is important to lead changes in organizational level. The participatory action research could be the effective method to drive organization changes. However, given the emphasis on social hierarchy in Asian culture, it is not easy to apply participatory action research programs, which requires critical thinking against an oppressive society [26]. Hence, it was not easy for the participants to request the company—where a hierarchical organization is stressed—for more job autonomy and less job burden for the sake of health, and they continued to work at night and over the weekend, resulting in little-to-no improvement in job stress. A meta-analytic review of the psychosocial environment in organizations and workers’ mental health [27] suggested that psychosocial change does not occur immediately but over a long period of time. Although the present study did not attempt to induce psychosocial change at an organization level, it can be suggested that participation not only of individual workers but also of the organization is important to promote workers’ health.

### 5.5. Sustaining Effect on Health-Promotion Behaviors

Although participant-centered programs are effective in sustaining health behavior in individuals, we could not find an existing study that examined how long after the health-promoting behavior would be maintained after the completion of participatory action research. Accordingly, the present study used survival analysis to predict whether participants would continue engaging in health-promoting behavior to reduce CVD risk. Survival analysis is a statistical approach which has been used in previous studies to predict the sustainability of nonsmoking habit [28] and of breastfeeding [29]. Additionally, LMM on the repeated measures of HPLP II was used to predict behavior change in the future based on pattern analysis of health-promoting behavior. It was predicted that the participants would continue practicing health-promoting behavior even after the completion of the study. This finding is consistent with a previous study, which showed that participatory action research is effective in sustaining healthy behavior. Although a previous study argued that the promotional activity is not associated with the prevention of professional risks [30], these results show that combining the two objectives, promotion and prevention, can be beneficial for the results of the proposed actions, especially of a participatory method is used. However, survival analysis only makes predictions based on a statistical method applied up to the time point of an event’s occurrence while taking into account partial information contained in the truncated data [28]. Accordingly, future studies should include a long-term follow-up period to confirm whether participants would continue practicing health-promoting behavior after the completion of the participant-centered intervention.

### 5.6. Factors Influencing the Sustainability of Health-Promotion Behaviors

To identify the factors influencing the sustainability of CVD-related health-promoting behavior in workers in small-to-medium sized workplaces, Cox regression analysis was performed, and CVD risk perception and self-regulation were identified as influential factors after controlling for covariates, such as gender, age, knowledge of CVD, social support, family function, and job stress. Currently, in the field of public health, health management is considered more important than disease management. For this reason, studies focusing on health promotion are being conducted, with an emphasis on self-regulation in sustaining health-promoting behaviors in individuals. However, it was difficult to find a study that objectively assessed improvement in self-regulation ability using participatory action research. Based on the objective data and multivariate analysis, such as Cox regression, the present study found that self-regulation indeed improves individuals’ health-promoting behavior. Therefore, this study provides objective data for future public health studies that will apply the participatory action paradigm to examine health behavior.

## 6. Conclusions

This action research study was conducted to reduce CVD risk in small-to-medium sized workplace employees classified as medically vulnerable. A health management journal was developed and used, which resulted in improved health-promoting behavior and self-regulation ability to continue practicing the behavior, as well as a reduction in CVD risk factors in workers. However, the small sample size (*n* = 34 including those who had left the company in the middle of the study) is not sufficient for a quantitative analysis. Furthermore, the study was conducted at a single, small-scale workplace located in a city, limiting the generalizability of the study findings to other workplace settings. Moreover, the observation period is too short to see all the changes. Nevertheless, the present study is significant because it proposes a methodological solution to reduce the risk of CVD in workers in small-to-medium sized workplaces and provides a momentum for raising policy interest by applying a participant-centered program and by testing its effectiveness on occupational nursing, a field that has not been actively researched. In the future, additional research should be conducted in other small-to-medium sized workplaces to validate the findings of the present study. In addition, a community-based participatory action research model encompassing organization, community, and policy levels should be developed and applied to maximize the effectiveness of participant-centered programs. In the future, longer follow-up studies should be conducted to more accurately identify the effects of participatory action research.

## Figures and Tables

**Figure 1 ijerph-16-04958-f001:**
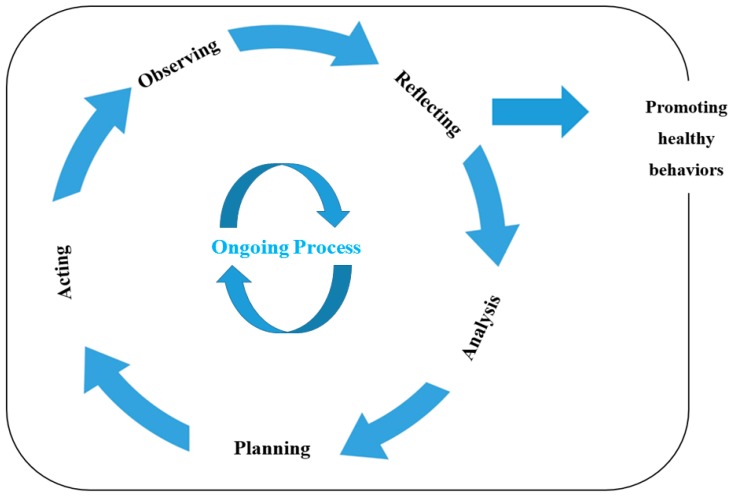
Conceptual framework of this study adopted from Lewin (1946)’s action research model.

**Figure 2 ijerph-16-04958-f002:**
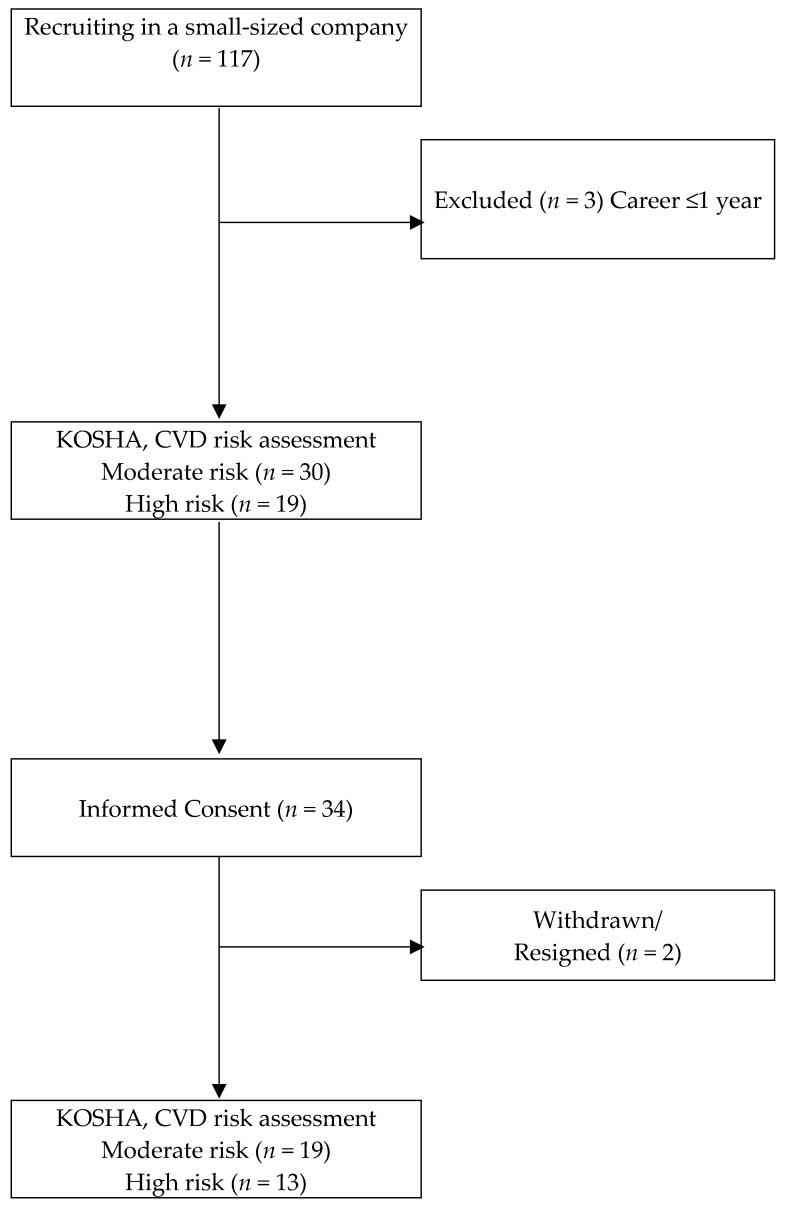
Algorithm Depicting Study Population. KOSHA: Korea Occupational Safety and Health Agency; CVD: Cardiovascular Disease.

**Figure 3 ijerph-16-04958-f003:**
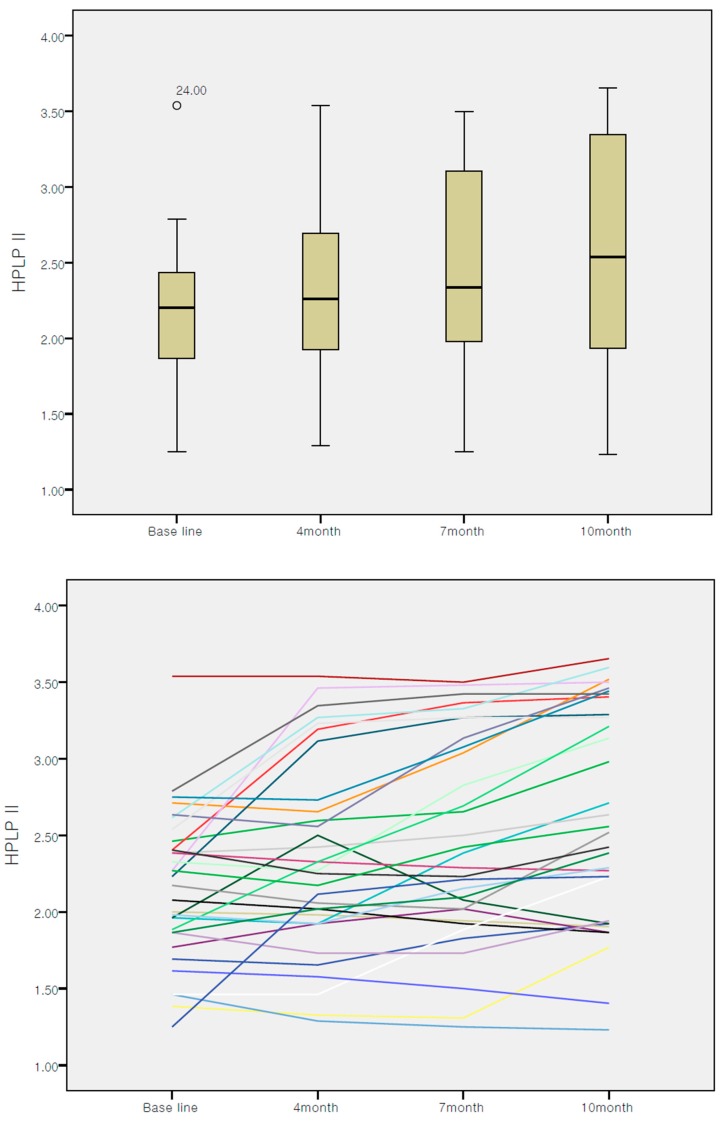
Changes of health-promotion behaviors Health-Promoting Lifestyle Profile II (HPLP II).

**Figure 4 ijerph-16-04958-f004:**
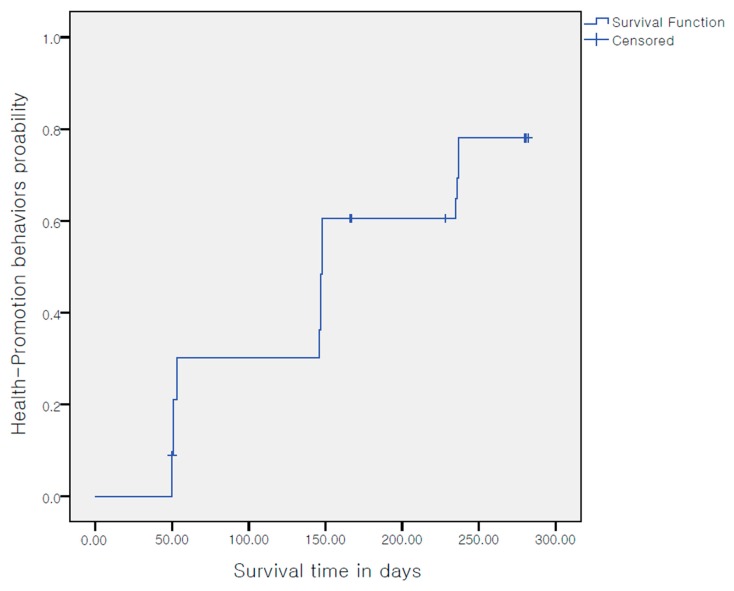
Survival function of health-promotion behavior.

**Table 1 ijerph-16-04958-t001:** Instruments used in this study.

Concepts	Instruments	Indication	Items
Self-Regulation (Exercise, Diet, Smoking, and Alcohol)	Treatment Self-Regulation Questionnaire (TSRQ)	Higher score	24
CVD risk perception	CVD Risk Perception	Higher score	5
Health-promoting behavior	Health-Promoting Lifestyle Profile II (HPLP II)	Higher score	52
Family Function	Family APGAR Questionnaire	Higher score	5
Social support	Interpersonal Support Evaluation List-12 (ISEL-12)	Higher score	12
Job stress	Effort-Reward Imbalance (ERI) + Overcommitment	Cutoff point = 1 (low/high)	23

CVD: Cardiovascular Disease; APGAR: Adaptation, Partnership, Growth, Affection, Resolve.

**Table 2 ijerph-16-04958-t002:** General characteristics of the study population (*N* = 32).

Variables	N (%)/Mean (SD)
Sex	
Male	26 (81.25)
Female	6 (18.75)
Age	42.54 (7.23)
20–39	11 (34.37)
40–59	21 (65.63)
Educational level	
≤High school	18 (56.25)
≥College	14 (43.75)
Marital status	
Unmarried	11 (34.37)
Married	21 (65.63)
Monthly income (10000 won)	
100~199	20 (62.50)
200~299	12 (37.50)
Working hours (/week)	55.61 (6.01)
Employment	
Regular	27 (84.47)
Contract	5 (15.53)
Job tenure (year)	5.25 (2.34)
<3	5 (15.62)
3–5	9 (28.13)
5–7	10 (31.25)
>7	8 (25.00)
Current smoking state	
Nonsmoker	14 (43.75)
Smoker	18 (56.25)

**Table 3 ijerph-16-04958-t003:** The results of action research to reduce CVD risk factors (*N* = 32).

Variables	Pre	Post	χ^2^ or t	*p*-Value
	Mean (SD)/*n* (%)	Mean (SD)/*n* (%)		
Body weight (kg)	77.22 (16.17)	75.31 (13.84)	2.97	0.006
Waist circumference (cm)	86.03 (11.84)	85.22 (9.78)	1.47	0.152
Systolic blood pressure (mmHg)	135.88 (13.39)	132.56 (10.51)	3.99	<0.001
Diastolic blood pressure (mmHg)	88.97 (8.36)	87.539 (7.07)	2.17	0.037
LDL cholesterol (mg/dL)	126.69 (48.69)	120.25 (41.46)	1.82	0.078
HDL cholesterol (mg/dL)	55.56 (32.02)	62.66 (30.93)	5.05	<0.001
Total cholesterol (mg/dL)	204.78 (44.97)	201.34 (38.41)	1.50	0.143
Triglyceride (mg/dL)	256.47 (220.50)	255.00 (215.60)	0.07	0.937
Fasting blood glucose (mg/dL)	109.50 (36.83)	102.63 (18.81)	1.65	0.110
Body Mass Index (kg/m^2^)	26.40 (5.16)	25.74 (4.30)	3.07	0.004
Normal weight	7 (21.8)	7 (21.9)	28.03	<0.001
Overweight	8 (25.0)	13 (40.6)		
Obesity	17 (53.2)	12 (37.5)		
Current smoking state				
Nonsmoker	14 (43.7)	21 (65.6)	3.09	0.066
Smoker	18 (56.3)	11 (34.4)		
CVD risk knowledge	11.94 (5.84)	12.25 ( 5.53)	1.89	0.067
Health-promoting behavior	2.16 (0.49)	2.62 ( 0.71)	5.82	<0.001
Self-regulation				
Exercise	16.37 (7.26)	20.21 (7.95)	3.44	0.002
Diet	18.31 (4.99)	23.37 (7.16)	3.58	0.001
Smoking	18.78 (5.39)	29.94 (7.30)	4.79	<0.001
Alcohol	14.28 (3.88)	14.31 (4.17)	0.19	0.845
CVD risk perception	9.91 (3.29)	8.50 (2.44)	3.23	0.003
Social support	22.56 (5.29)	22.09 (4.77)	520	0.607
Family function	3.63 (2.41)	6.03 (1.47)	6.98	0.001
Job stress	1.06 (0.34)	1.16 (0.29)	3.25	0.003
CVD risk groups				
Low risk	0 (0.0)	7 (21.9)	8.83 ^†^	0.008
Moderate risk	19 (59.4)	18 (56.3)		
High risk	13 (40.6)	7 (21.9)		

^†^ Fisher’s exact test; LDL: Low-density lipoprotein; HDL: High-density lipoprotein.

**Table 4 ijerph-16-04958-t004:** Factors affecting health-promotion behavior maintenance.

Variables	Hazard Ratio	95% Confident Interval	*p*-Value
Age	1.06	0.96 to 1.16	0.203
Sex	0.45	0.13 to 1.58	0.217
CVD risk perception	4.03	1.29 to 12.60	0.016
CVD knowledge	0.93	0.86 to 1.01	0.092
Self-regulation	1.05	1.02 to 1.09	0.002
Social support	1.11	0.99 to 1.24	0.068
Family function	0.92	0.75 to 1.13	0.455
Job stress	0.55	0.11 to 2.57	0.450

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
