# Peer review of "Developing a Health-Promotion Program Based on the Action Research Paradigm to Reduce Cardiovascular Disease Risk Factors among Blue Collar Workers"

_ijerph, 2019, doi:10.3390/ijerph16244958_

Round 1

Reviewer 1 Report

this study intends to propose a program for the promotion of cardiovascular health suitable for small and medium enterprises. The study adopts a participatory method, and this is certainly advantageous, as indicated by numerous literature reviews.

The promotional activity is not associated with the prevention of professional risks. There are indications that combining the two objectives, promotion and prevention, can be beneficial for the results of the proposed actions, especially if a participatory method is used [Magnavita N. Obstacles and Future Prospects: Considerations on Health Promotion Activities for Older Workers in Europe. Int J Environ Res Public Health. 2018 May 28;15(6). pii: E1096. doi: 10.3390/ijerph15061096.]. The authors could discuss this point.

The study was well designed and conducted. The main problem is the small number of participants.

Minor changes:

Line 203. in the ERI questionnaire the cut-off does not divide stress from the absence of stress, but only the perception of high stress from that of low stress. The statement “An ER-ratio less than 1 indicates absence of job stress and an ER-ratio greater 1 indicated the presence of job stress” should be modified.

Author Response

Point 1: The promotional activity is not associated with the prevention of professional risks. There are indications that combining the two objectives, promotion and prevention, can be beneficial for the results of the proposed actions, especially if a participatory method is used [Magnavita N. Obstacles and Future Prospects: Considerations on Health Promotion Activities for Older Workers in Europe. Int J Environ Res Public Health. 2018 May 28;15(6). pii: E1096. doi: 10.3390/ijerph15061096.]. The authors could discuss this point.

Response 1: Thank you for the constructive comments. We inserted the sentences about the potential effect of combining the two objectives in discussion part.

Point 2: Line 203. in the ERI questionnaire the cut-off does not divide stress from the absence of stress, but only the perception of high stress from that of low stress. The statement “An ER-ratio less than 1 indicates absence of job stress and an ER-ratio greater 1 indicated the presence of job stress” should be modified.

Response 2: We modified the sentences about the ERI questionnaire.

Reviewer 2 Report

Introduction Very long description but it does not make clear why the selected research model is better than others that exist in the literature. Method Justifying their participation in the 34 surveys, 117 candidates were selected and only 34 have participated is unclear because this low response rate. 129 A small-to-medium sized business enterprise with 117 workers, located in S city, was selected. (errors in text) Results It would be necessary to some explanatory table the data are convoluted and are difficult to understand by this reviewer. The results should be clearer and more concise, and structured for easy understanding. Discussion They should include practical implications and look for more up-to-date appointments. The number of up-to-date appointments in the discussion is minimal. Bibliography: There is a lot of obsolete literature (> 5 years)

Author Response

Response to Reviewer 2 Comments

Point 1: Introduction Very long description but it does not make clear why the selected research model is better than others that exist in the literature.

Response 1: We modified introduction to make clear.

Point 2: Method Justifying their participation in the 34 surveys, 117 candidates were selected and only 34 have participated is unclear because this low response rate. 129 A small-to-medium sized business enterprise with 117 workers, located in S city, was selected. (errors in text).

Response 2: Thank you for your constructive comment. We modified the sentence what you pointed. Also, we inserted figure 2 to help readers understand the process of selecting study population.

Point 3: It would be necessary to some explanatory table the data are convoluted and are difficult to understand by this reviewer. The results should be clearer and more concise, and structured for easy understanding.

Response 3: We thank the reviewer for a thorough reading of the manuscript and valuable comments. We modified table 2 and described in more detail in results section.

Point 4: Discussion They should include practical implications and look for more up-to-date appointments. The number of up-to-date appointments in the discussion is minimal.

Response 4: We modified discussion to make clear. Also, except for references that must be cited, discussions were based on recent studies.

Point 5: Bibliography: There is a lot of obsolete literature (> 5 years)

Response 5: The references were reviewed once again, and unnecessary references were deleted.

Reviewer 3 Report

The manuscript describes an interesting program to promote cardiovascular health in a small number of workers.

The observation period is probably too short to see all the changes that a promotion program of this type can cause.

A longer period of observation would also allow us to assess the stability of the improvements achieved.

Minor changes:

L.195 The ERI questionnaire in its larger version (23 items) also contains 6 items on over-commitment or intrinsic stress

Author Response

Response to Reviewer 3 Comments

Point 1: The manuscript describes an interesting program to promote cardiovascular health in a small number of workers. The observation period is probably too short to see all the changes that a promotion program of this type can cause. A longer period of observation would also allow us to assess the stability of the improvements achieved.

Response 1: We thank the reviewer for a thorough reading of the manuscript and valuable comments. All authors agree with your opinion that the observation period is probably too short to see all the changes. Therefore, these explanations were included in the manuscript as one of limitations.

Point 2: Minor changes:

L.195 The ERI questionnaire in its larger version (23 items) also contains 6 items on over-commitment or intrinsic stress.

Response 2: We modified the sentence what you pointed.

Reviewer 4 Report

The Article “Developing a Health Promotion Program Based on  the Action Research Paradigm to Reduce  Cardiovascular Disease Risk Factors among Blue Collar Workers”  is very clear and well written. I greatly appreciated the longitudinal study with repeated measures (both medical parameters and self report psychological measurement). Discussions and conclusions are also adequate.

I suggest creating a summary table with all the measures used (n. Items, range of response, dimensions of the psychometric scales, medical parameters) to facilitate the reader.

Graphically improve table 2 (pre-post test) because it is not very clear (I suggest not using the symbol ± to divide mean and SD). Values ​​that have changed significantly could be bold.

Finally in table there are some written in Korean (top right)  that should be translated.

Author Response

Response to Reviewer 4 Comments

Point 1: I suggest creating a summary table with all the measures used (n. Items, range of response, dimensions of the psychometric scales, medical parameters) to facilitate the reader.

Response 1: Thank you for your constructive comment. We inserted a summary table (table 1) with the all measurements used in this study.

Point 2: Graphically improve table 2 (pre-post test) because it is not very clear (I suggest not using the symbol ± to divide mean and SD). Values ​​that have changed significantly could be bold.

Response 2: We deleted the symbol ± used in table2. Also, numbers with significant changes have been changed to bold.

Point 3: Finally in table there are some written in Korean (top right) that should be translated.

Response 3: Thank you for your comment. We deleted Korean in Figure 4.

Round 2

Reviewer 2 Report

The changes seem right to me.